# Comparative and Predictive Analysis of Electrical Consumption during Pre- and Pandemic Periods: Case Study for Romanian Universities

**Paul Cristian Andrei** [1], **Marilena Stanculescu** [1], **Horia Andrei** [2,*], **Ion Caciula** [2], **Emil Diaconu** [2], **Nicu Bizon** [3], **Alin Gheorghita Mazare** [3], **Laurentiu Mihai Ionescu** [3] and **Marian Gaiceanu** [4]

1   Department of Electrical Engineering, University Politehnica Bucharest, 060042 Bucharest, Romania
2   Faculty of Electrical Engineering, Electronics and Information Technology, University "Valahia" of Targoviste, 130004 Targoviste, Romania
3   Faculty of Electronics, Communications and Computers, University of Pitesti, 110040 Pitesti, Romania
4   Faculty of Control Systems, Computers Science, Electrical Engineering and Electronics, University "Dunarea de Jos" Galati, 800008 Galati, Romania
*   Correspondence: horia.andrei@valahia.ro or hr_andrei@yahoo.com; Tel.: +40-729-104-202

**Abstract:** The pandemic period was caused by COVID-19 and it has been an unprecedented event in the last 100 years of human history. Regarding universities, major changes have occurred both in the online method of education as well as in the patterns of their electrical consumption, respective of both students' and teachers' residential electrical consumption. The focus of this research is to conduct and assess a comparative analysis of universities' electrical consumption during the pre- and pandemic periods. Polynomial regression is used to model the electrical consumption of four Romanian universities during the period 2019–2021. Also, this study proposes a method for predicting the electrical consumption of universities in three months of 2021, compared to that of the same months in 2020. The data analysis shows that the electrical consumption had decreased between 20.6% and 36.29% in the pandemic period of 2020 compared to that 2019. Additionally, this study evaluates the electrical consumption of universities due to their use of computers, which represents an important percentage of the total consumption; this was between 11.28% and 60.5% in the pre-pandemic year 2019, but this was substantially reduced in 2020, to be between 57.13% and 77.27%. Based on the data that has been provided by students and teachers, the calculated values show that the electrical residential consumption increased by about 20 kWh per month and per computer unit during the pandemic.

**Keywords:** electrical consumption; universities; pre- and pandemic period; online and face-to-face education; polynomial regression; prediction; residential consumption

## 1. Introduction

Among the important goals of today's society are the increase in the quality and efficient use of electricity. The in-situ monitoring of electrical consumption patterns represents the measure of these desideratums, and they are the initial data from which any analysis starts. According to the European Parliament's Directive 2012/27/EU on energy efficiency, the strategic goal of each country, in terms of their national energy policy, is the security of the energy supply through sustainable and competitive development, but with the saving of primary energy resources, the reduction of pollutants that have a greenhouse effect, and the decrease in energy consumption of 19% by 2020 as presented in [1,2].

The World Health Organization has declared that the spread of COVID-19 has been a global pandemic since March 2020 [3]. Almost all countries in the world have taken urgent measures to limit the spread of the SARS-CoV-2 virus, such as working from home, instating quarantine and restricting travel. These restrictive measures continued

in 2021, with different intensities, depending on the severity of the national outbreaks. Unfortunately, the pandemic caused the loss of many lives, and it undoubtedly disrupted the normal life, business and economy of the countries of the world [4]. The deadlock in business and the significant decline in global trade led to a global recession in 2020, with the global gross domestic product (GDP) falling by 4.4%, as shown in [5]. During the pandemic, in 2020, the international energy market showed various fluctuations that were caused by COVID-19, and the global demand for electricity decreased by 5%, which led to a positive impact on the environment. Also, the data that were collected from more than 30 countries showed a 27% increase in electrical consumption in public, industrial and commercial buildings as a result of the fact that more and more people were forced to work and learn from home, as is shown in [6]. The restrictive measures that were imposed by the pandemic and their duration were the main factors that affected energy demand. The International Energy Agency (IEA) has developed interesting studies that address both the impact of the COVID pandemic on the global energy demand in relation to carbon emissions [7], and a review of global energy consumption [8]. According to these studies, after in a month of being in a lockdown in 2020, the demand for electricity decreased by 20% when it was compared to that of the previous year, 2019, and the average decrease was 1.5% for the whole year. Then, in June 2020, the mitigation of these isolation measures led to a demand for electricity that was only 10% lower than it was in the same month of 2019, and it decreased in July by 5% less than the level was in the same month of the previous year, in countries such as the United Kingdom, India, Spain and France. In EU countries, the electricity demand started to recover to levels that were close to those that were reached in 2019 in August, and then the demand fell steadily in the following months as more restrictive measures were reintroduced. Globally, electricity supply fell by 2.6% in the first quarter of 2020 when it is compared to that of the same period in 2019, while electricity production from renewable sources increased by 3%, which is explained by the new investments that were made in renewable energy systems in 2020.

At the same time, the pandemic has had an unprecedented impact on global education, as stated in the study that was published by the United Nations [9]. In more than 190 countries, the measure of closing schools and universities was introduced, which affected over 94% of the number of children and students in the world. In a short time, the educational, pre-university and academic societies adapted to the new conditions and migrated from an interactive and practical face-to-face system to a predominantly online and virtual digital environment, as presented in [10,11]. Visits to the virtual world have become a regular routine and as a natural consequence, whiteboards and notebooks have been replaced by laptops, tablets and computers. Children and students have had to adapt immediately to online learning sessions, being connected daily to a digital screen to participate in lessons, courses, seminars, laboratories sessions and other complementary activities. Obviously, the digital information and communication technology skills of both teachers and students have increased, but at the same time, the gaps in digital skills and resources have created inequalities between different educational institutions at national and global levels.

An important change that has been generated by the face-to-face activities of these institutions is related to the decrease in electrical consumption, which was constant in all of the universities in the world. For example, European universities have had decreases of 10% to 40%, as shown in [12]. Several papers have analyzed the impact that the shutting down of the universities' physical activities during the COVID-19 pandemic had on the energy consumption of the universities' buildings. An analysis of the electricity use and its economic impacts for buildings with electric heating under the lockdown conditions is presented in [13]. The authors of this study chose, as calculation examples, the educational buildings and residential buildings in Norway. The occupant density in the buildings is one parameter which affects the energy consumption and virus transmission risk in buildings, alike. In this respect, the authors of article [14] propose the optimum occupant distribution patterns that account for the lowest number of infected people and the minimum amount of energy consumption taking, as an example to study, a university building. The main

objective of article [15] was to analyze the impact of the COVID-19 lockdown on the energy consumption of university buildings, taking into account the climate adjustments to the baseline period conditions. The authors apply their methodology to 83 academic buildings of the Polytechnic University of Catalonia—Barcelona Tech, Spain. The measured data showed a substantial decrease in energy consumption in the academic buildings. A similar aim was developed in article [16]. Based on pre- and pandemic period measured data, the authors analyzed the impact that closing the campus and several buildings in University of Almeria, Spain has on energy consumption. Some energy saving measures were proposed to achieve the minimum amount of waste of the energy. Another research study assessed the unregulated electrical consumption within a single laboratory building which functions as a research engineering building in the higher education sector, during the COVID-19 lockdown [17]. The study showed that the electricity consumption typically reduced during the lockdown period with a percentage of 46.61% when this was compared with that of the period before the lockdown. In terms of the environmental impact, the authors of study [18] analyzed and compared the carbon footprint that a mid-sized UK University produced during the COVID-19 lockdown against that which was generated within the respective time period in previous years. The results show that the overall carbon footprint of the UK University decreased by almost 30% during the lockdown. In [19], the authors evaluated the effects of the COVID-19 pandemic on electrical consumption in 13 state universities in Michoacán, Mexico. Through a comparative assessment of their electrical energy consumption before and during pandemic, the authors of this study estimated a reduction in energy consumption and its economic and environmental beneficial impacts in the presence of the COVID-19 pandemic.

On the other hand, during the pandemic, when many activities, including educational ones, moved to the online environment, household electrical consumption increased. Several researchers have focused on this important topic. Thus, the authors of study [20] analyzed the increase in residential building electrical consumption during the COVID-19 pandemic. The measured data prove that the highest percent increases in non-HVAC residential loads occurred between 10 a.m. and 5 p.m. Another study focused on the identification of the impacts that staying home living patterns had on the energy consumption of residential buildings [21]. Measured data that were collected from various reported sources were reviewed and analyzed to assess the changes in the overall electricity demand for various countries and US states. The analysis results indicate that the energy consumption for the housing sector has increased by as much as 30% during the full 2020 lockdown period.

As a major consumer of electricity for universities, the campuses have a different consumption pattern than that of residential buildings because they must consider heating and ventilation systems, lighting systems, the number of computers, laboratory equipment and last, but not least, the occupancy rate with teaching, administrative and student staff. Theoretically, the electricity consumption of academic buildings should be zero when they are empty. However, in order to ensure the minimum operation of the basic activities that can be performed remotely, it is necessary for the servers, databases, other computer systems, emergency lights, elevators and security systems to function, which implies that they perform a minimum amount of energy consumption, which increases with the number of these devices, even in the case of the depopulation of campuses.

In Romania, the entire public sector consumes about 10% of the final energy amount and 25% of this energy is represented by electricity [22]. Public schools are important consumers of electricity. Thus, for example, in Romania the total number of public educational institutions, of which, 5% are universities, represents about 25% of the total number of public institutions [23]. This distribution is shown in Figure 1. Both public and private higher education universities generally have the same irregular footprint (pattern) of electrical consumption, due to the existing electrical equipment and systems in the laboratories, courses and application rooms, buildings for administrative staff, campuses and in dormitories and canteens, and to all of this is the addition of the important electrical consumption of modernization and maintenance works [24].

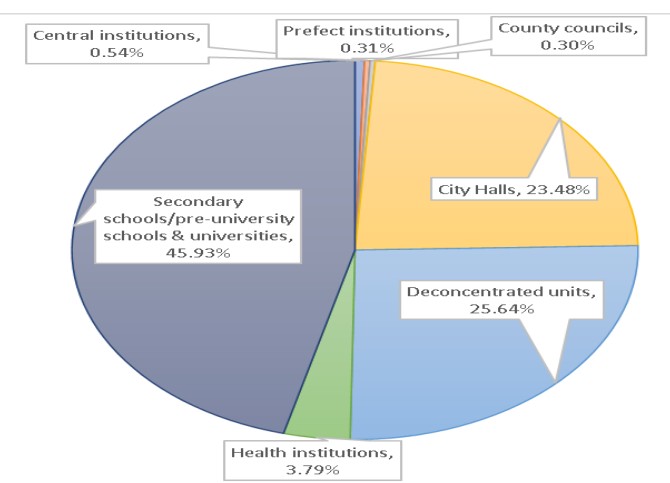

**Figure 1.** Distribution of public institutions in Romania.

In Romania, the higher education institutions have been developed mainly in urban areas where there are more important consumers of electricity and where the expansion of the microgrid will change the paradigm of the evolution of electricity networks in future smart cities. That is why the monitoring and analysis of electrical consumption in universities is becoming an important topic [25].

During the pandemic, when worldwide education—including higher level education—moved to the online system, there were significant changes in electricity consumption. In the context of the worrying spread of the COVID-19 virus, on 16 March 2020, the President of Romania signed the decree on establishing a state of emergency on the territory of the country, which meant that all educational institutions in Romania, including universities, had to suspend their courses [26]. This decree came into force after the Romanian Government had previously announced the lockdown between 11 and 22 March of all educational units, including universities [27]. Thus, starting from 11 March 2020, about 380,000 students and over 30,000 university teachers in Romania had to deal with the online education system, a situation that would last until the first semester of the 2021 academic year and, partially, even during the second semester [28].

In order to cope with these new teaching and learning methods and techniques, both teachers and students had to intensively use computers and tablets, especially at home. In this regard, the governments of the world states took urgent measures to provide computers to all those who were involved in the online education process. It was found that the number of computers was insufficient for all students and teachers in order to carry out online education to a high standard. For this reason, the Romanian Government decided on and announced, in May 2020, the purchase and distribution of 250,000 tablets for students and teachers to facilitate their distance learning activities. For university education, the transition from face-to-face to online education has been both a challenge to accommodate teachers and students with this new type of education and a significant change in the consumption of electricity by universities and households for staff, teachers and students [29].

In most of the previously cited articles, quantitative and detailed analyses have been used to evaluate the changes in electrical consumption in universities and residential households during the pre- and pandemic periods. In a past work, authors have analyzed the electrical consumption of four Romanian universities, which have different consumption patterns in the pre- and pandemic periods [30]. Also, a comparative analysis of the monthly electrical consumption in the pre- and pandemic periods of the studied universities, which are located in the capital and in three other important cities, is performed, based on the data that were collected from monthly measurements. Several new contributions have been developed in this study when it is compared to other publications in the field. For example, in this study, the electrical consumption of four Romanian universities has been modeled

using polynomial regression. In this regard, the Python programming environment is used, and the obtained approximation polynomials had a good accuracy. Another software application is implemented to generate the polynomial models for the months of March–May of the 2020 pandemic period, which are compared to the real values of the same months from 2021. The results that are obtained using this predictive method were close to the measured values. Additionally, based on the data that were collected from each university, such as the number of computers and the average monthly consumption, the authors calculated the percentage of electrical consumption due to the computers from the universities' total consumption in the pre- and pandemic periods. During the lockdown of 2020, minimum operating levels for the computers were required to ensure that the core activities of the universities could be performed remotely, such as the use of the servers, access to email, databases or software tools, which led to a substantial reduction in electrical consumption. Consequently, a significant part of the universities' consumption has been transferred to residential electrical consumption through the intensive use of personal computers. Based on the data that has been provided by several teachers and students of the four studied universities for the pre- and pandemic periods, the authors analyzed the growth of the residential electrical consumption levels during the pandemic. We emphasize that the data were used only for families whose energy footprints (number of people, housing area, and important household consumers) did not change between the two studied periods.

This research is organized into four sections. Section 2 presents the structure of the four studied universities in Romania and the steps of the proposed methodology. Section 3 describes the polynomial models for electrical consumption of universities during the pre- and pandemic periods, presents the prediction of electrical consumption during three months of 2021, based on the mathematical consumption model from 2020, and evaluates the electrical consumption due to use of computers and the transfer of this consumption from universities to residential households during the pandemic. The conclusions are drawn in Section 4 to highlight the changes in the electrical consumption of universities and residential households during the pandemic as compared to those of the pre-pandemic period.

## 2. Materials and Methods

In the first part of this section, the general information about the four studied Romanian universities is presented. It includes historical data and the structure of the universities, the number of teachers and students, the number of students living in dormitories and the number of computers in the university. All these data are used in the integrated analysis of the electricity consumption before and during the pandemic, in Section 3. Finally, in this section a step-by-step methodology of the proposed model is described.

The University "Valahia" of Targoviste (UVT) is one of the youngest state universities in the Romanian academic world. It was founded in 1992 and had contained in its structure, 2 faculties, a university college, 14 specializations and approximately 7500 students [31]. With it being the 30th anniversary of the founding of the university, UVT finds that it is completely adapted to the new requirements of the business environment and also to the current social environment. Due to its continuous development, today, UVT has in its structure 10 faculties with 35 specializations, master's and doctoral studies, a department of teacher training and open distance learning as well as an Institute of Scientific Research and Multidisciplinary Technology (ICSTM). From the point of view of the number of students that were attending the UVT courses in the academic year 2021–2022, approximately 7500 were enrolled, of which almost 400 had their accommodation in the 3 student dormitories that are located inside the university campus. UVT's human resources are composed of approximately 600 employees, of which 330 are teachers and approximately 270 are research staff, non-teaching and administrative assistants. All of them operate in lecture and seminar rooms, laboratories and project rooms, as well as administrative offices. For the completion of teaching and administrative activities, the university has a number of approximately 1000 computers and laptops. In order to calculate the total energy consumption of the computers/laptops on the university campus in the pre-pandemic period, one

can also add the approximately 400 personal computers/laptops that students have in their dormitories, where they are accommodated. Thus, in the pre-pandemic period, about 1400 computers were connected to the university's electricity network. At the time of the declaration of the state of emergency, since all the didactic activities were carried out online, the number of computers that remained connected to the electrical network decreased to approximately 350 units. The reason for this is that only the computers of the administrative and research staff as well as of a small percentage of students (approximately 60 students) who remained accommodated for extraordinary reasons (they had jobs or were involved in research projects) did function in the university. Otherwise, all the teaching staff and the rest of the students worked/studied from home, exclusively online.

The Polytechnic University of Bucharest (UPB) is the oldest and most prestigious engineering school in Romania, with a tradition that has accumulated in its 200 years of existence, and this defines its uniqueness by creating knowledge through research and technological innovation, as well as by its implementation through vocational education and training at the European level [32]. Currently, UPB has 15 faculties and provides for the over 30,000 students, a high quality of education, which is supported by complex research activity, in accordance with the requirements and with the means that are offered by the modern information society. UPB organizes, through its faculties:

(a) undergraduate studies, which are organized around 95 study programs in 17 scientific fields;

(b) master's degree studies, which are structured around 22 scientific fields, totaling a number of 188 master's degree programs (32 study programs in foreign languages);

(c) doctoral studies, which are organized around 14 scientific fields in the doctoral schools; the total number of Ph.D. students from UPB increased in the period 2016–2021, and so currently, 1521 Ph.D. students are registered, which represents a share of about 5.4% of all UPB students;

(d) continuous training university studies.

The UPB community consists of 1300 teachers and researchers and a number of approximately 30,000 students who are enrolled in various study programs: bachelor's, master's or doctorate (of which, there are over 1000 foreign students) working in the 53 departments of the 15 faculties and in over 40 research centers. Their activity is supported by the auxiliary teaching staff which totals 972 people, as well as by over 480 administrative staff members. The research areas in which the academic and research staff of UPB (academic staff) perform are engineering sciences, applied sciences (mathematics, physics, chemistry and computer science), economics and social sciences, education and human sciences (philology—foreign languages). UPB has a university campus that includes, for each faculty, for each study program, for each research activity and for the administration, the necessary spaces so that the activities are carried out in accordance with the national and international standards. UPB has the dormitory–canteen complexes, Regie and Leu, that contain 28 dormitories and a capacity of 13,046 seats and 6 canteens.

The University of Pitesti (UPIT) is located in the central-southern part of Romania, on the upper course of the Arges River, in the Muntenia region, Arges County. It was the first higher education school to be established in the city of Pitești in 1962 under the name of "3-year Pedagogical Institute" [33]. Its main goal was to offer pedagogical training in 6 fields of study: philology, biology, physics, chemistry, mathematics and physical education. Moreover, in 1966, after the Pitesti Automobile Plant Dacia-Renault was set up, technical specializations were established which led to the establishment of the Institute of Engineers. In 1974, as a result of the development of technical education, "The Institute of Higher Education" was established. For a long time, it was subordinated to the Bucharest Polytechnic Institute. Nonetheless, starting from the 22 March 1991, by the Ministerial Order 4894, the Institute became the current "University of Pitesti" (UPIT). In these years, both the dynamics and the economic development of the Muntenia region and the European context have influenced the development of the study programs, as well as their diversity in the University. Therefore, the University of Pitesti (UPIT) has 6 faculties,

2 research centers, 20 departments, 46 bachelor's programs, 41 master's degree programs and 4 doctoral schools, covering 10 doctoral fields. In all of these study programs, there are over 5780 undergraduate students, 2350 master's students and around 135 doctoral students who are enrolled, totaling over 8300 students. For the students' training, UPIT has 3 University Campuses, a sports base with 2 gyms and 3 outdoor fields. In addition, it has 2 student dormitories that can accommodate up to 700 students and 3 dining locations.

The teaching and research activity is supported by 365 teaching staff, 20 researchers, and the support activity is provided by 135 auxiliary teachers and 62 non-teaching staff. In its teaching and research spaces, the University has as significant consumers of electricity, with over 2200 desktops or laptop computers. Moreover, in the dormitories, there are around 500 computers, the majority of which being laptops. In addition, there are around 4 CNC machines, and there are 6 lathes to which are added to the lighting installations of the course rooms, seminar classrooms, laboratories, halls and the lighting installations of the gyms which have a capacity of 22 kW. For example, if in the pre-pandemic period the average electric power consumption was 62 kW/month, in the pandemic period, in total lockdown, the monthly average electric power consumption decreased to 40 kW/month, due to remote working practices (online teaching activities). Furthermore, during 2021, when those teaching activities became partially practiced online, the monthly average electric power consumption increased at 56 kW/month, and this increase is determined by the laboratory activities, research, and the return of the students to the gyms and to the dormitories.

University "Dunarea de Jos" is located in the city of Galati, in the southeastern part of Romania. The existence of higher education facilities in Galați began in 1948 with the establishment of the Institute of Land Improvements, the first faculty in the country with this profile [29]. Higher education in Galați appeared as a result of the approaches of the economic and cultural institutions of the city and the county, being closely correlated with the Danube River, which crosses the city of Galați. Thus, the development of higher education in Galați has focused on a series of unique study programs in the country: shipbuilding, ship and port operation, food industry, fishing equipment and refrigeration, which have materialized in various forms of higher education, culminating in their merging and the creation of the University of Galați, in 1974 (Decree of the State Council of 20 March 1974). The current name, University "Dunărea de Jos" of Galați (UDJG), dates from 1991 and was instated by the Order of the Minister of Education and Science (Government Decision of 4 January 1991). UDJG assumes the mission of generating and transferring knowledge to society through the initial and continuous training of its citizens at university and postgraduate level, in order for them to achieve personal development, the professional realization of the individual and to satisfy the competency need of the socio-economic environment through scientific research, development, innovation and technological transfer through individual and collective creations that are made in the field of science, engineering, economics, arts, socio-human sciences, medical sciences and legal sciences, and by ensuring the performance and physical and sports development of these people, as well as capitalizing on and disseminating their results. Currently, within the UDJG there are 14 faculties that prepare students for undergraduate, master's and doctoral studies in various fields (e.g., technical, socio-human, economic, artistic, health, etc.). UDJG is the most important higher education institution in southeastern Romania, with a total of about 12,500 students in recent years. The doctoral studies are carried out in 16 doctoral fields, under the coordination of 111 doctoral supervisors. In terms of the number of students that attended UDJG courses in the academic year 2021–2022 there were enrolled, approximately 12,238 students, of which almost 3000 had accommodation in the 7 student dormitories that are located inside the university campus. The human resources departments of UDJG are composed of approximately 2200 employees, of which 1200 are teachers and approximately 1000 are research staff, non-teaching and administrative assistants. As is the case at UVT, the human resources staff work in lecture and seminar rooms, laboratories and project rooms as well as administrative offices. For

the completion of teaching and administrative activities, the university has a number of approximately 3810 computers and laptops. In order to calculate the total consumption of computers/laptops on the university campus in the pre-pandemic period, we must also add the approximately 3000 personal computers/laptops that students have in their dormitories, where they are accommodated. Thus, in the pre-pandemic period, about 7000 computers were connected to the university's electrical network. At the time of the declaration of the state of emergency, since all the didactic activities were carried out online, the number of computers that remaining connected to the electrical network decreased to approximately 2500 units. The justification for this decrease was to carry out the activity online, leaving a small number of students or researchers in the dormitories who were involved in research projects.

The methodology that is used in this study to estimate the impact of coronavirus lockdown on the electrical consumption of universities and residential households is presented synthetically in the flowchart that is in Figure 2.

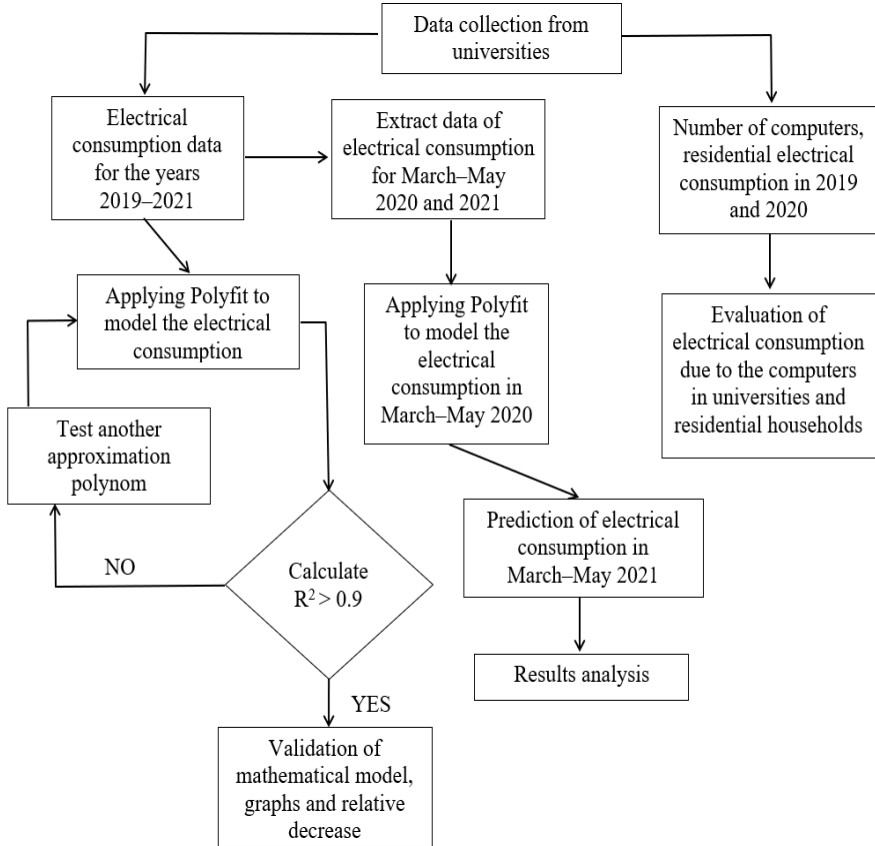

**Figure 2.** Methodology flowchart.

Furthermore, the steps that constitute this methodology are explained in detail in the following text:

*Step 1.* For the years 2019–2021, the following data were collected and tabulated from the four Romanian universities—UVT, UPB, UPIT, and UDJG: electrical consumption; number of teachers; total number of students and number of students living on campus; number of computers; average number of hours computers worked in the pre-pandemic period; number of computers used while working in the pandemic; the average consumption of a computer; the residential electrical consumption of teachers and students during pre- and pandemic periods. Note that the data were collected from teachers' and students' families considering that they had the same energy footprints, i.e., the same number of people, housing area, and important household consumers in the two studied periods.

*Step 2*. The implementation of polynomial regression by using the "polyfit" function of the NumPy Python programing library was performed to obtain the polynomial models of the electrical consumption of the studied universities during the period from 2019–2021. The approximation polynomials were selected and validated if their *R*-square ($R^2$) values were bigger than 0.9. Graphical characteristics, which put in the evidence of the real and modeled data of the electrical consumption and the $R^2$ values, are presented.

*Step 3*. The polynomial model of electrical consumption—defined as the predicted values—from the months March–May in 2020 was generated. In addition, a new Python application extracted—using a similar "polyfit" function as is described in step 1—the data that was related to the months of March–May in 2020 and built the "9 degree" polynomial model (predicted model) of electricity consumption for these three months. Then, these predicted values were compared graphically with the real data from the same months of 2021. The prediction for these three months was chosen because, in this period, in the four studied universities, the educational system of practice was carried out online. Since the amount of data that were used were very few (only 3), the accuracy of the method was defined simply by analyzing the metric distances between the predicted and the measured values.

*Step 4*. The calculation of the electrical consumption due to the use of computers and the estimation of the average residential electrical consumption of teachers and students for each university in the pre- and pandemic periods were made. For this purpose, many data from universities, teachers and students were used.

*Step 5*. The interpretation and analysis of the results that were obtained in Steps 2, 3 and 4 included the polynomial models, the predictive analysis and the transfer of electrical consumption from universities to residential households.

In previous studies [34,35], the authors have used machine learning algorithms such as SVM, ANM, LASSO, LR, GB or RF for predicting cost and gas emissions in integrated energy–water optimization models in buildings, respectively, to forecast the levels of greenhouse gas emissions that will be produced. Based on large data collections, the proposed algorithms provided the results with accurate prediction. Unlike these articles, the current study is based on a small number of electrical consumption data from each university, totaling only 12 values for each year. Therefore, polynomial regression was chosen for the mathematical modeling of the annual electrical consumption of each university, which finally provided very precise results.

The monthly measurement values of the four universities' electrical consumption were performed with data from the years: 2019—before the pandemic with normal educational activity, 2020—during the pandemic period with a lockdown, and 2021—normal and lockdown educational activity constituted the input data for the software applications. In the PyCharm Community environment of the Python 3.10.5 programing language, the applications of polynomial regression by a "polyfit" function have been created to construct graphs of the real data and the polynomial electrical consumption models for each university for the years 2019–2021 [36]. For determining, with the good accuracy, the polynomial approximations of the electrical consumption for each university, a Python script was developed to compute the $R^2$.

To examine the accuracy of the polynomial regression procedure, the most used indicator was the *R*-square ($R^2$) which determined the correlation between the measured (observed) values and the approximated (predicted) values. For the *N* acquired values, one was denoted by $x_i$ the measured values were denoted by $1 \le i \le N$, and denoted by $y_i$ were the predicted values, and this was conducted using a numerical or statistical method. Then, one can compute $R^2$ using the following formula [37]:

$$R^2 = 1 - \frac{\sum\limits_{i=1}^{N} (x_i - y_i)^2}{\sum\limits_{i=1}^{N} (x_i - \overline{x})^2} \tag{1}$$

where $\bar{x}$ represents the arithmetic mean of the $x_i$ values. A value of $R^2 = 1$ means that the numerical model correctly predicted the measured values. A value of $R^2$ towards 0 means that prediction model was wrong.

Considering the number of computers and their average working time for each university during the pre- and pandemic periods, the calculation of their corresponding electrical consumption was performed for the years 2019–2021. Also, based on a non-public investigation that was conducted on a significant number of teachers and students at each university, it was possible to determine an average increase in residential consumption due to the intensive use of personal computers during the pandemic. In this research, only the data referring to household consumption which kept the same consumption footprint (number of people, housing area, and important household consumers) in before and during the pandemic were taken into account. Thus, the energy transfer from the reduced electrical consumption of the universities to the increased residential consumption was estimated.

## 3. Results and Discussions

All of the results of the proposed numerical methodology and the discussions considering the specifics of electrical consumption irregularity for each university are presented below.

### 3.1. UVT: Electrical Consumption Results and Interpretation

The measurement data for the monthly electrical consumption of UVT for the years 2019–2021 are presented in black in the graphs in Figures 3–5.

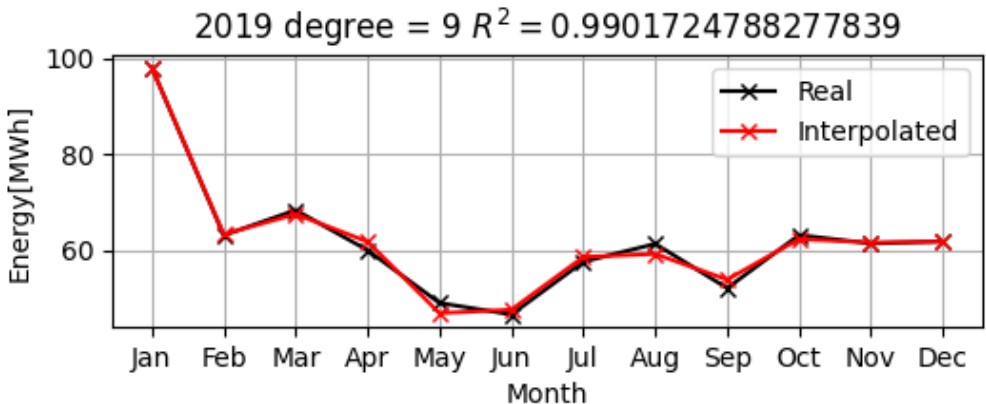

**Figure 3.** Data and polynomial model of electrical consumption of UVT during 2019.

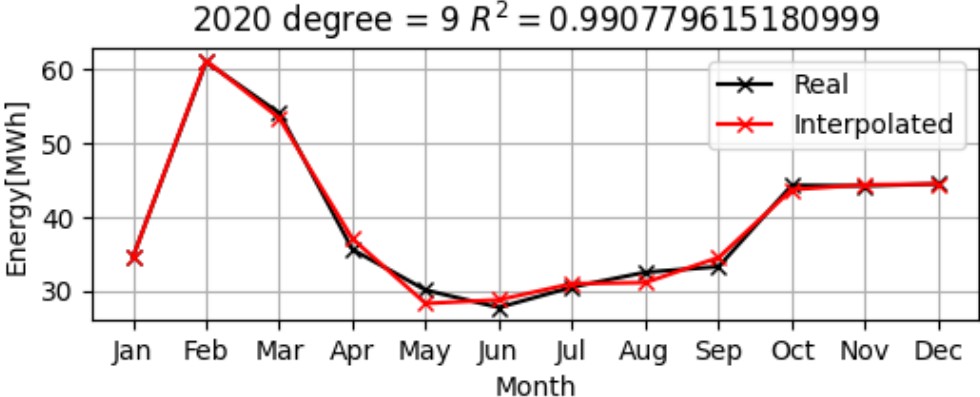

**Figure 4.** Data and polynomial model of electrical consumption of UVT during 2020.

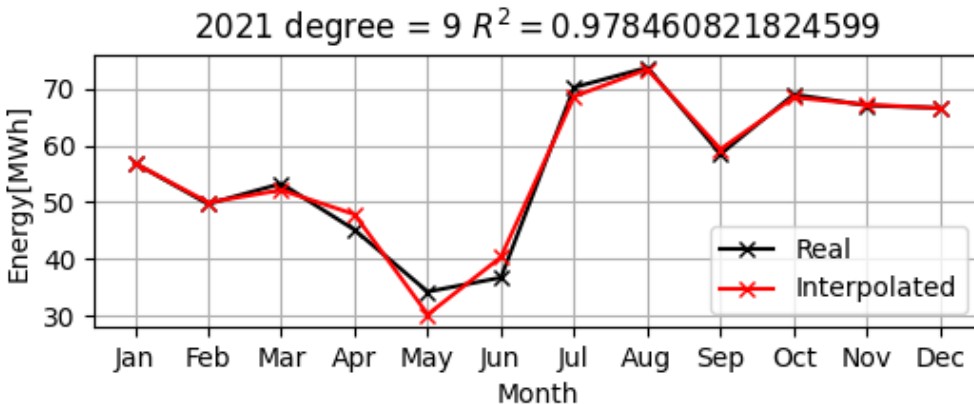

**Figure 5.** Data and polynomial model of electrical consumption of UVT during 2021.

The polynomial regression function "polyfit" from the NumPy library has been used to determine, as accurately as possible, the models for electrical consumption in the three studied years, which are illustrated in red in the graphs from Figures 3–5. The polynomial approximations of the electrical consumption for the years 2019–2021 are of degree 9. For example, the polynomial model of the total electrical consumption of UVT in 2020 is represented by:

$$y(\text{UVT,2020}) = 4.92 \times 10^{-5} \times x^9 - 2.51 \times 10^{-3} \times x^8 + 5.29 \times 10^{-2} \times x^7 - 5.96 \times 10^{-1} \times x^6 + 3.92 \times x^5 + 1.59 \times 10^1 \times x^4 + 4.55 \times 10^1 \times x^3 - 1.08 \times 10^3 \times x^2 + 1.83 \times 10^2 \times x - 7.31 \times 10^1 \tag{2}$$

where $y$ represents the annual electrical consumption in MWh and $x$ is the monthly consumption.

Also, a Python script application was developed to compute the $R^2$ and to choose the best accuracy for the mathematical model. Thereby, the approximation polynomials of degree 9 provide the best accuracy for the model, such that the calculated values of coefficient $R^2$ are 0.97 and 0.99, as is presented in Table 1.

**Table 1.** Polynomial models of electrical consumption for UVT, UPB, UPIT and UDJG during the period from 2019–2021.

| Year | University | Degree of Approximation Polynomial | $R^2$ |
|------|------------|-----------------------------------|-------|
| 2019 | UVT | 9 | 0.99 |
|      | UPB | 9 | 0.91 |
|      | UPIT | 9 | 0.92 |
|      | UDJG | 9 | 0.96 |
| 2020 | UVT | 9 | 0.99 |
|      | UPB | 9 | 0.98 |
|      | UPIT | 9 | 0.90 |
|      | UDJG | 9 | 0.99 |
| 2021 | UVT | 9 | 0.97 |
|      | UPB | 9 | 0.59 |
|      | UPIT | 9 | 0.96 |
|      | UDJG | 9 | 0.99 |

The analysis of the data that was obtained from the measurements shows that there was a decrease in the total annual electrical consumption from 2020 and 2021 during the pandemic, compared to 2019, which was taken as a reference university year with face-to-face education practices.

Table 2 shows the total annual consumption of UVT in the three studied years and the relative percentages of their consumption reduction when they are compared to those of 2019. The year 2020 was an academic year with online education only which led, at UVT, to

a significant reduction in their energy consumption. It is worth noting that in 2021, a mixed education took place, featuring both online and face-to-face practices, which led to a lower percentage decrease in the electrical consumption when it is compared to that of 2019.

**Table 2.** Total annual (for the years 2019–2021) electrical consumption and its relative decrease during the pandemic when the figures are compared to that of 2019 at UVT, UPB, UPIT and UDJG.

| Year | University | Total Annual Electrical Consumption (MWh) | Relative Decrease Compared to 2019 (%) |
|---|---|---|---|
| 2019 | UVT | 742.078 | - |
| | UPB | 21,631 | - |
| | UPIT | 723.885 | - |
| | UDJG | 1113.01 | - |
| 2020 | UVT | 472.729 | 36.29 |
| | UPB | 17,164 | 20.6 |
| | UPIT | 532.709 | 26.41 |
| | UDJG | 712.18 | 20.6 |
| 2021 | UVT | 680.794 | 8.25 |
| | UPB | 18,634 | 13.85 |
| | UPIT | 669.999 | 7.44 |
| | UDJG | 1453.95 | −30.63 |

From March–May in 2020, the university education took place online and there were no exam sessions. Then, in a similar way, during the same months of March–May in 2021, the education also took place online. Therefore, considering the analogy of these periods during the two years of online education—2020 and 2021—a prediction of electrical consumption was made. By using the "polyfit" function, the polynomial (predicted) model of electrical consumption was created for March–May in 2020 and it was compared with the real electrical consumption of the same period in 2021. In Figure 6, the monthly electrical consumption of the predicted model from March–May in 2020 is represented in red, whereas the real data from the same period in 2021 is in black.

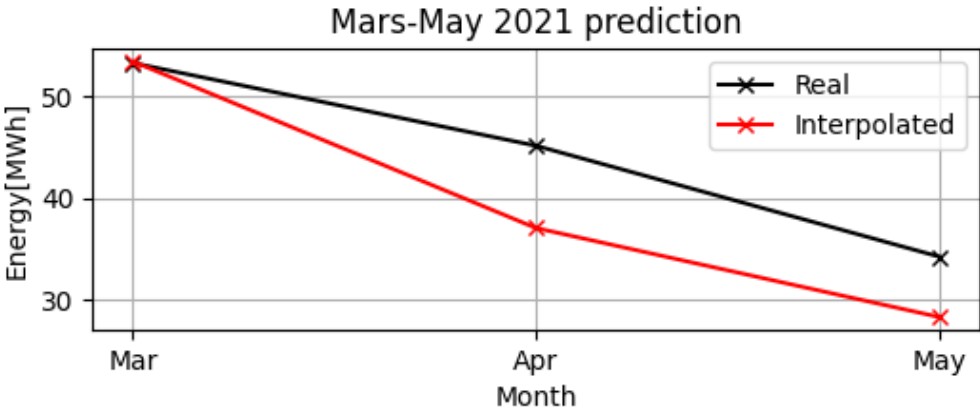

**Figure 6.** Predicted model and real electrical consumption for Months March–May in 2021 for UVT.

Considering that only three values were used for the polynomial regression, the calculation of the coefficient $R^2$ is no longer required to verify the accuracy of the predicted model. Although the trends start at the same point (they have the same electrical consumption in March), for the measurements at the middle of March, the distance between the real values and the predicted ones begins to grow, which proves the irregularity of electrical consumption in university. Thus, in 2021 there was an increase of distance between the real and predicted values of electrical consumption of 7.5 MWh in April and of 6 MWh in May, respectively.

Obviously, a part of the reason for the reduction in electrical consumption is due to the significant decrease in the number of computers/laptops that operated in the university and in the students' campus. As presented in Section 2, it is considered that in the period before the pandemic, in 2019, 1400 computers were operating in UVT, with an average consumption of 130 Wh. During the pandemic period of the 2020 academic year, when during the entire academic year, the didactic activity took place online, 350 computers operated, meaning that there were 1050 fewer units. These 350 units ensured the uninterrupted operation of the administrative services and the ongoing research projects. Taking into account an average operating time of 8 h/day within 185 days per year, it results in the total annual electricity consumption of 269.36 MWh in 2019 and 67.34 MWh in 2020. From the total electrical consumption data of UVT that are presented in Table 2, the consumption due to the operation of computers represents 36.29% in 2019, which is a respectively lower percentage than the 14.24% in 2020. If 2019 is taken as a reference for the total annual consumption of computer units, then in the pandemic period of 2020, the consumption due to the operation of computers decreased by 75% as is presented, synthetically, in Table 3. The consumption of computer units in 2021 was not taken into account either, because this is the year that the university education took place both online and face-to-face.

**Table 3.** Number of utilized computers and electrical consumption due to computers in 2019 and 2020 at UVT, UPB, UPIT and UDJG.

| Year | University | Total Number of Utilized Computers | Relative Decrease of Utilized Computers Compared to 2019 (%) | Electrical Consumption due to Computers from Total Annual Consumption (%) |
|------|-----------|-----------|-----------|-----------|
| 2019 | UVT | 1400 | - | 36.29 |
| | UPB | 11,000 | - | 11.28 |
| | UPIT | 2700 | - | 26 |
| | UDJG | 7000 | - | 60.5 |
| 2020 | UVT | 350 | 75 | 14.24 |
| | UPB | 2500 | 77.27 | 3.23 |
| | UPIT | 635 | 76.48 | 10 |
| | UDJG | 250 | 96.42 | 3.37 |

This decrease in the electrical consumption due to the operation of computers in UVT, will be transferred to the corresponding increase in the household consumption of teachers and students. Thus, a nonpublic survey that was conducted on a significant number of teachers and students whose household electrical energy footprints (number of people, housing area, and important household consumers) did not change substantially in 2019 and 2020 showed an average monthly electrical consumption of 130 kWh/month in 2019 and of 150 kWh/month in 2020. It was also found that compared to the 1 h/day before the pandemic (2019), the computer activity in the homes of teachers and students increased to about 9 h/day during the pandemic (2020), i.e., this increased by 8 h. Taking into account 300 days/year, for a 130 Wh computer it results a higher electrical consumption during 2020 when this is compared to 2019, by 312 kWh/year or by 26 kWh/month. So, this approximative calculation of the increase in monthly household electrical consumption in 2020 when it is compared to that of 2019, 130 kWh + 26 kWh = 156 kWh, is very close to the one that results from the study (150 kWh), with the error being $\varepsilon = 3.8\%$.

### 3.2. UPB: Electrical Consumption Results and Interpretation

The measurement data for the monthly electrical consumption of UPB for the years 2019–2021 are presented in black in the graphs, respectively, in Figures 7–9.

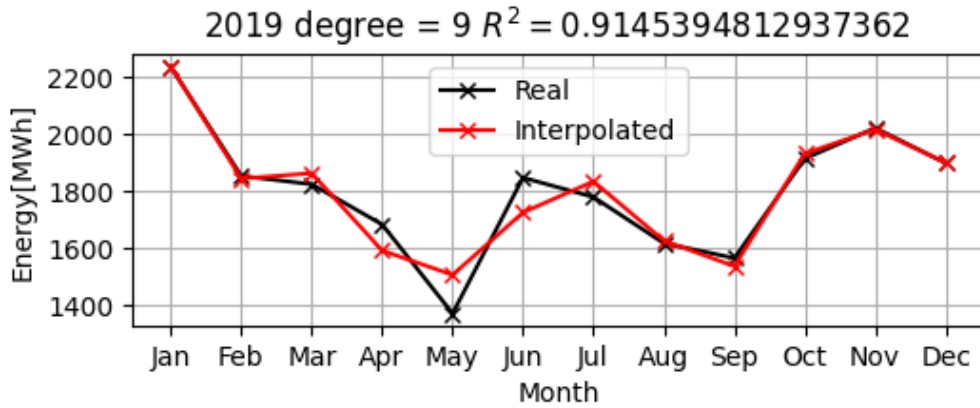

**Figure 7.** Data and polynomial model of electrical consumption of UPB during 2019.

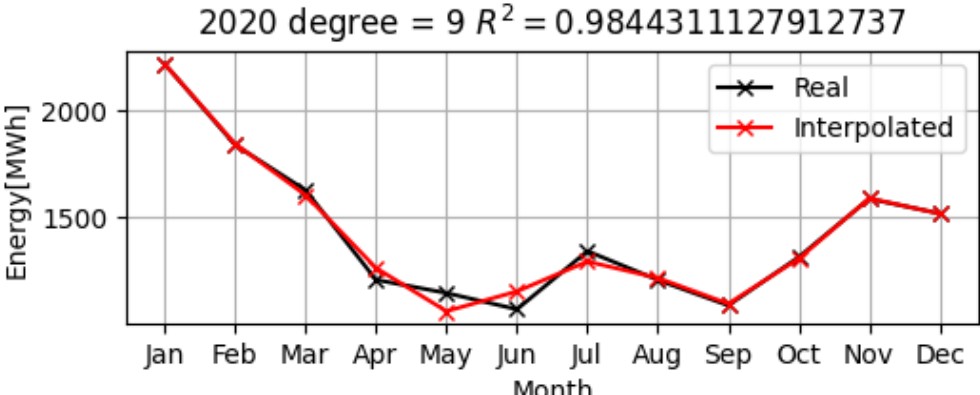

**Figure 8.** Data and polynomial model of electrical consumption of UPB during 2020.

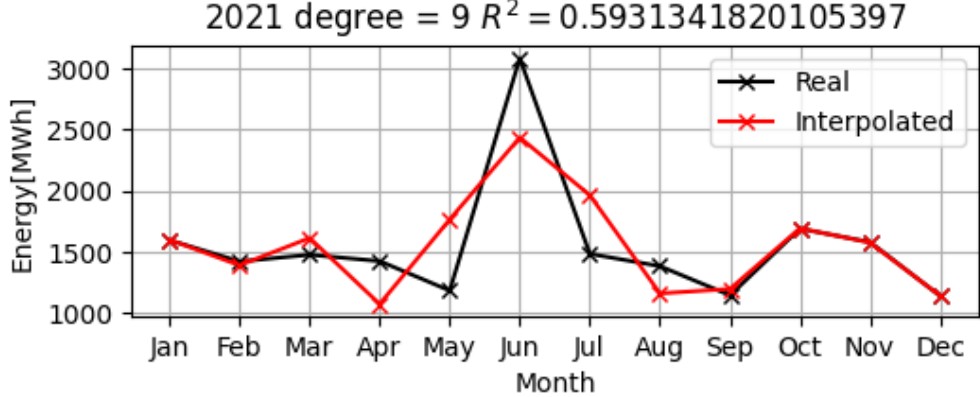

**Figure 9.** Data and polynomial model of electrical consumption of UPB during 2021.

The same polynomial regression application as was used for UVT was used here. The determination of the models of electrical consumption in the three studied years, are illustrated in red in the graphs in Figures 7–9, respectively. The polynomial approximations of the electrical consumption for the years 2019–2021 are of degree 9. For example, the polynomial model of the total electrical consumption of UPB in 2020 is:

$$y(\text{UPB,2020}) = 2.68 \times 10^{-3} \times x^9 - 1.47 \times 10^{-1} \times x^8 + 3.4 \times x^7 - 4.24 \times 10^1 \times x^6 + 3.1 \times 10^2 \times x^5 - 1.36 \times 10^3 \times x^4 + 3.5 \times 10^3 \times x^3 - 4.96 \times 10^3 \times x^2 + 3.05 \times 10^3 \times x + 1.7 \times 10^3 \tag{3}$$

where $y$ represents the annual electrical consumption in MWh and $x$ is the monthly consumption.

The approximation polynomials of degree 9 that are provided for the years 2019 and 2020 represent the best accuracy for the model, such that the calculated values of coefficient

$R^2$ are 0.91 and 0.98, respectively. Due to the very high electricity consumption in June 2021 because of important modernization, maintenance and repair works in the UPB campus, the optimal value of coefficient $R^2$ is 0.59 as shown in Table 1.

Table 2 shows the total annual consumption of UPB in the three studied years and the relative percentages of their consumption reduction when compared to that of 2019. The year 2020 was only one with solely online education which led, at UPB, to a decrease in the electrical consumption of 20.6%. It is noteworthy that in 2021 this decrease was 13.85%. Electrical consumption in 2021 decreased, comparatively, to the figures of 2019 but they increased when they were compared to that of 2020. The increase was due, on one hand, to the development of mixed education, which involved both online and face-to-face practices, but also to the fact that UPB had carried out a series of works regarding the rehabilitation of the university campus.

Figure 10 shows the monthly electrical consumption polynomial model from March–May in 2020 in red, respective to the real electrical consumption in the same period in 2021 in black. The trends start at different points in March, wherein the predicted value is smaller than that of the real one. Near the middle of March, the two trends intersect, and then the real electrical consumption values become higher than the predicted ones do, reaching a difference of 219 MWh in April and of 148 MWh in May.

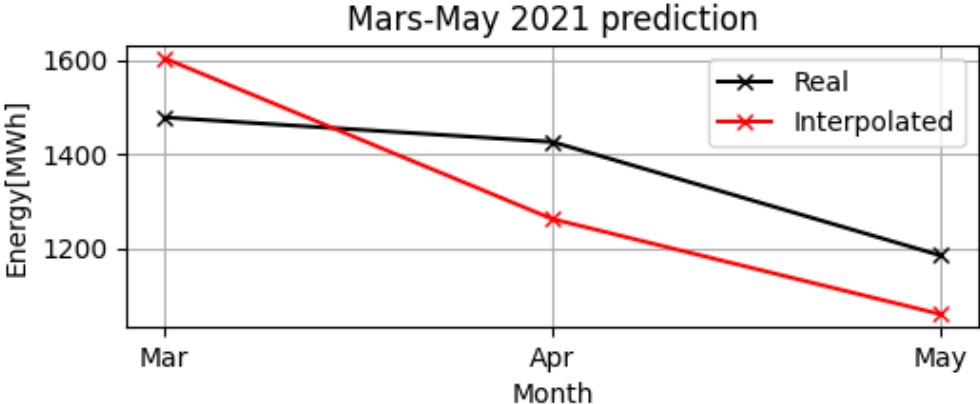

**Figure 10.** Predicted model and real electrical consumption for Months March–May in 2021 for UPB.

During 2019, in UPB, an estimated total number of 11,000 computers were assigned to administrative services, research centers, courses, seminaries and laboratories rooms and the university library had been functioning with an average consumption of 150 Wh/computer. Taking into account that each computer was used on average 8 h/day for about 185 working days, it results an annual consumption of 2442 MWh, which means that there was an 11.28% decrease in the total electricity consumption in 2019. The decrease of electricity consumption in 2020 was mainly due to the high reduction in the number of computers that were functioning inside the university campus and the students' dormitories.

During 2020, when online education replaced face-to-face education, the number of functioning computers were reduced to about 2500, representing the workstations and servers that were assigned to the research centers and administrative services. Taking into consideration the same functioning values as those for 2019, that gives a total amount of consumed energy of 555 MWh, which means that there was 3.23% of the total energy consumption in 2020. This shows that the energy that was consumed by the computers decreased in 2020 by 77.27% compared to that in 2019 as is presented in Table 3.

On one hand, the transition to online education in 2020 led to the diminishing of the total of university energy consumption, but on the other hand, the household consumption of teachers and students increased. In 2021, as seen in Table 2, a slight increase of 1470 MWh or 8.56% can be seen in the total energy consumption when it is compared to that of 2020, which is partly due to the introduction of mixed education.

Only a percentage of the decrease in the electrical consumption that is due to the use of computers in UPB has been transferred to the increase that is seen in household the electrical consumption of teachers and students. Thus, a nonpublic study that was conducted on a significant number of teachers and students whose household energy footprints (number of people, housing area, and important household consumers) did not change substantially in 2019 and 2020 showed an average monthly electrical consumption of 138 kWh/month in 2019 and of 165 kWh/month in 2020. It was also found that compared to the 1 h/day before the pandemic (2019), the computer activity in the homes of teachers and students increased to about 9 h/day during the pandemic (2020), i.e., this increased by 8 h. Taking into account 300 days/year, for a 130 Wh computer, this results a higher electrical consumption during 2020 compared to that of 2019, by 324 kWh/year or by 27 kWh/month.

### 3.3. UPIT: Electrical Consumption Results and Interpretation

Below are the measurement data for UPIT's monthly electrical consumption for 2019–2021, shown in black lines in Figures 11–13, respectively.

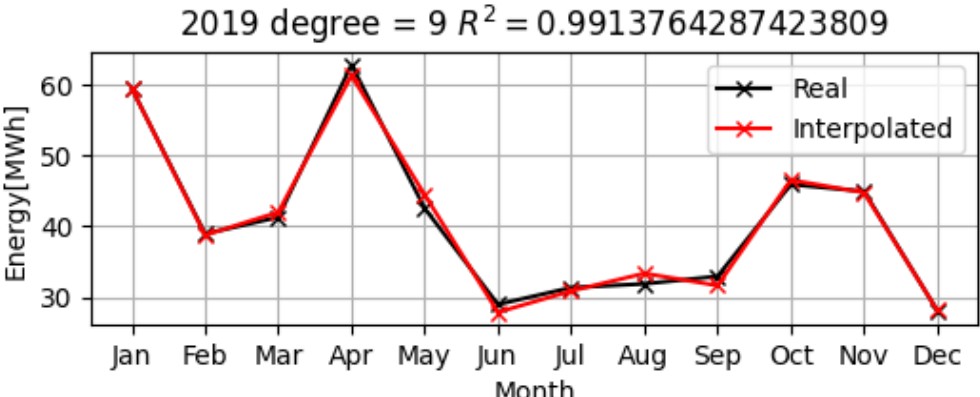

**Figure 11.** Data and polynomial model of electrical consumption of UPIT during 2019.

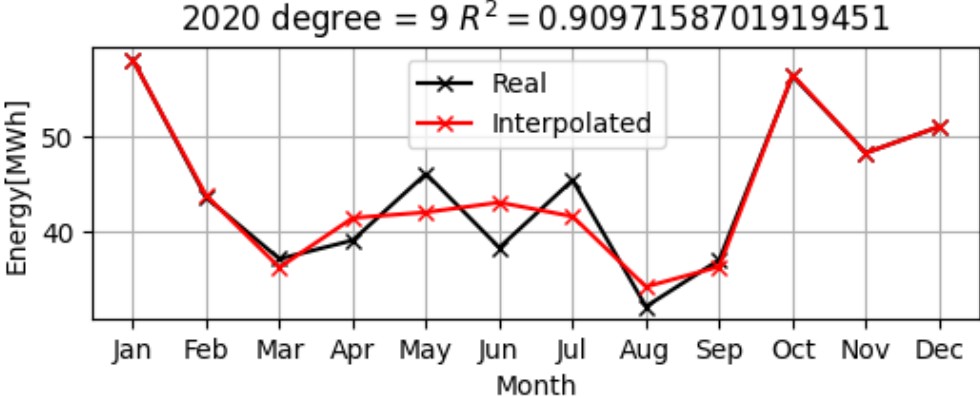

**Figure 12.** Data and polynomial model of electrical consumption of UPIT during 2020.

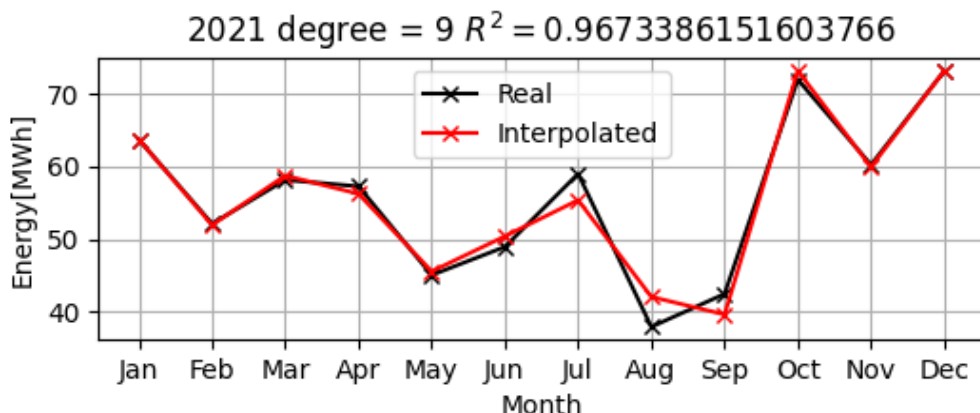

**Figure 13.** Data and polynomial model of electrical consumption of UPIT during 2021.

As can be seen in Figures 11–13, the polynomial model (represented by the red graphs) is quite close to the real data for the electrical consumption over the three years, 2019–2021. For 2019, the residual sum of the squares between the real values and those obtained from the model ($SS_{res}$) was approximately 55.44, while the total sum of the squares ($SS_{tot}$) was 790.89. For the year 2020, it was $SS_{res} = 63.92$ for an $SS_{tot} = 707.99$, and for the year 2021, it was $SS_{res} = 44.08$ while $SS_{tot} = 1349.70$. The polynomial approximations of the electrical consumption for the years 2019–2021 are of degree 9. For example, the polynomial model of total electrical consumption of UPIT in 2021 is:

$$y(UPIT,2021) = 5.58 \times 10^{-4} \times x^9 - 3.1 \times 10^{-2} \times x^8 + 7.28 \times 10^{-1} \times x^7 - 9.41 \times x^6 + 7.33 \times 10^1 \times x^5 - 3.53 \times 10^2 \times x^4 + 1.04 \times 10^3 \times x^3 - 1.78 \times 10^3 \times x^2 + 1.58 \times 10^3 \times x - 4.9 \times 10^2 \tag{4}$$

where $y$ represents the annual electrical consumption in MWh and $x$ is the monthly consumption.

The approximation polynomials of degree 9 provide the best accuracy for the model, such that the calculated values of coefficient $R^2$ are, respectively, 0.90, 0.92 and 0.96, as is presented in Table 1.

In the case of UPIT, the analysis of the data that were obtained by measurements shows a decrease in the total electricity consumption in the period 2020–2021 compared to that of 2019, as illustrated in Table 2. In 2020, the educational activity was exclusively online, so the consumption decreased by 26.41% when it was compared to that of 2019, which reflects the difference in electrical consumption between the situations in which the studies are taken with an online presence and an on-site presence. In 2021, online and hybrid activity took place depending on the evolution of the pandemic in the region. The hybrid activity (over an interval of about 3/4 of the total activity period) involved the development of laboratory and project activities among the students in a half-group mode, so this meant a that there was a doubling of the number of laboratory activities that were carried out. The courses and seminaries were conducted online. All this is reflected in a 7.44% reduction in the energy consumption when it is compared to that of 2019.

Figure 14 shows the polynomial model of electrical consumption from March–May in 2020 in red, and the real consumption in the same period of 2021 is represented in black. Although the two trends start in March at different points where the value of the metric distance, 20.97 MWh, is at its maximum, they tend toward gaining close electrical consumption values. Thus, in April, the distance between the real values and the predicted ones decreases to 18.14 MWh, and then, starting from the middle of April, the distance between the values decrease, reaching a minimum value 1.08 MWh in May.

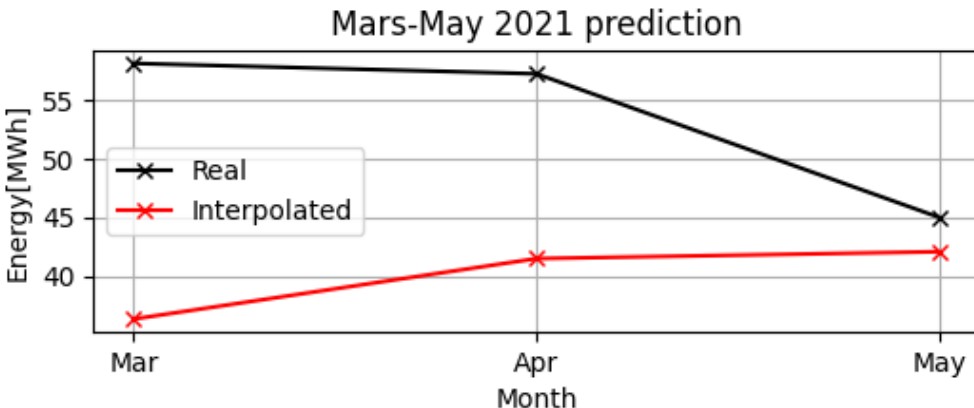

**Figure 14.** Predicted model and real electrical consumption for Months March–May in 2021 for UPIT.

The reduction in electrical consumption due to the operation of desktop computers or laptops inside the university and the students' hostel is significant. During 2019, in UPIT, around 1900 desktop computers with an average consumption of 160 Wh (120 Wh central unit + 40 Wh monitor) and about 800 laptops with an average consumption of 110 Wh were used. Given as input data were the number of computers and the average energy consumption of each of them, but also included was the fact that a computer is used on average for 8 h/working day, 185 working days per year, and has an annual energy consumption of 226 MWh, which represents about 26% of the total electrical consumption. During the pandemic period in 2020, the number of operational computers was reduced to about approximately 635, representing the workstations and servers that were assigned to the research centers and administrative services. The electrical consumption level that was reached due to the use of these computers was 53.2 MWh which represent 10% of the total annual electrical consumption in 2020.

In 2021, the energy consumption that was due to the operation of the computer units in UPIT increased with the partial physical return of students in the university, and the annual energy consumption increased by 77 MWh, compared to that of 2020, which represents an increase in 41% and a level of 17% of the total 723.8 MWh electrical consumption of UPIT.

Based on a nonpublic survey that was conducted on a significant number of teachers and students of UPIT whose household energy footprints in 2020 did not change substantially when they were compared to those of 2019, the collected data showed an average monthly electrical consumption of 128 kWh/month in 2019 and of 144 kWh/month in 2020. It was also showed that compared to an average of 1 h/day before the pandemic (2019), the computer activity in the homes of teachers and students increased to an average of 9 h/day during the pandemic (2020), i.e., a difference of 8 h/day. Taking into account 300 days/year, for a 130 Wh computer, it results a higher electrical consumption during 2020 compared to that in 2019, by 312 kWh/year or by 26 kWh/month.

### 3.4. UDJG: Electrical Consumption Results and Interpretation

Below are the data of the measurements for the monthly electrical consumption of UDJG for the years 2019–2021, represented in black, as well as the obtained polynomial models of electrical consumption shown in red, in Figures 15–17.

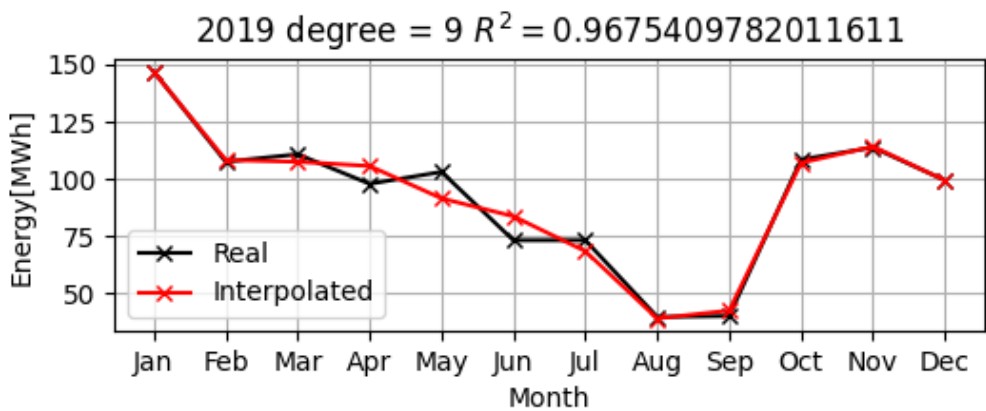

**Figure 15.** Data and polynomial model of electrical consumption of UDJG during 2019.

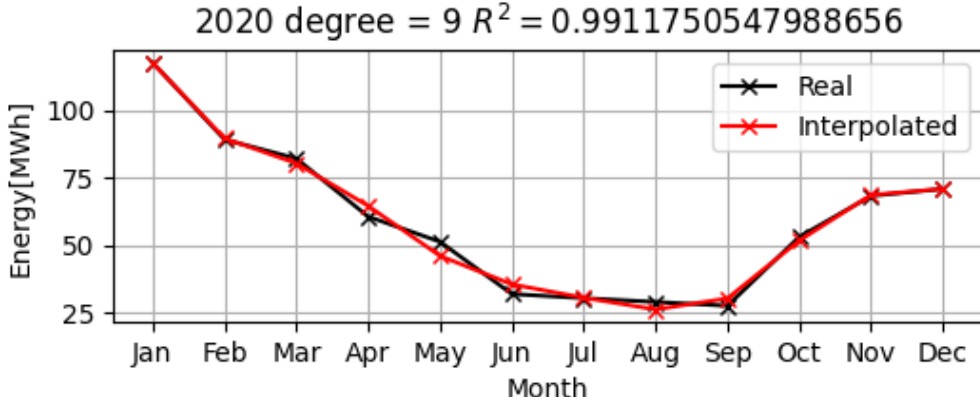

**Figure 16.** Data and polynomial model of electrical consumption of UDJG during 2020.

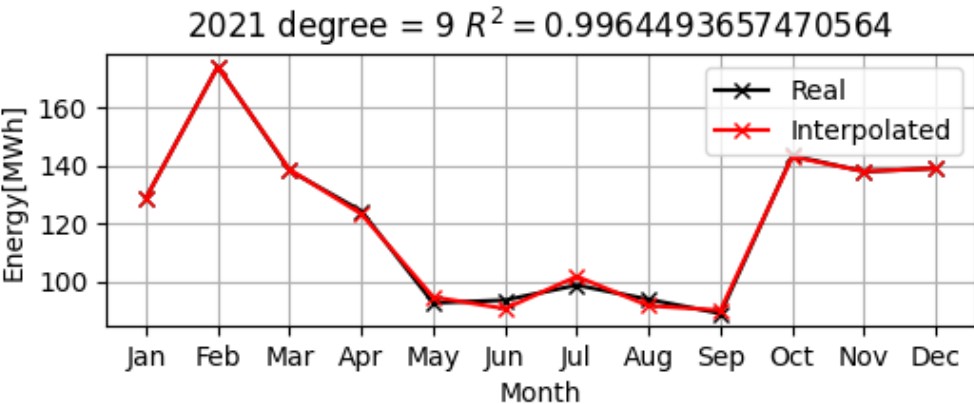

**Figure 17.** Data and polynomial model of electrical consumption of UDJG during 2021.

As can be seen in Figures 15–17, the polynomial model (represented by the red lines on the graphs) is quite close to the real data for electrical consumption over the three years, 2019–2021. The values of $R^2$ that are greater than 0.9, i.e., they are 0.96 and 0.99, respectively, have been calculated for all of the years, which demonstrates that the approximation by the nine-degree polynomial model is very accurate, as shown in Table 1.

For example, the polynomial approximation that was obtained using numerical regression for the total electrical consumption in 2019 can be expressed by:

$$
\begin{aligned}
y(\text{UDJG},2019) = {} & 6.2 \times 10^{-4} \times x^9 - 3.45 \times 10^{-2} \times x^8 + 8.1 \times 10^{-1} \times x^7 - 1.04 \times 10^1 \times x^6 + 8.13 \times 10^1 \times x^5 - 3.92 \times 10^2 \times x^4 \\
& + 1.15 \times 10^3 \times x^3 - 2 \times 10^3 \times x^2 + 1.7 \times 10^3 \times x - 4.30 \times 10^2
\end{aligned}
\tag{5}
$$

where $y$ represents the annual electrical consumption in MWh and $x$ is the monthly consumption.

Based on measured data from March–May in 2020, the "polyfit" function was used to build the prediction of the electrical consumption which is represented by the red characteristic in Figure 18. Also, the measured data for the same months, March–May, in 2021 are represented in black. The two trends start in March at different points where the value of the metric distance was 48.3 MWh. In April, the distance between the real values and the predicted values kept constant to 48.3 MWh, and then, starting from the middle of April, the distance values decrease, reaching a minimum value 38.87 MWh in May.

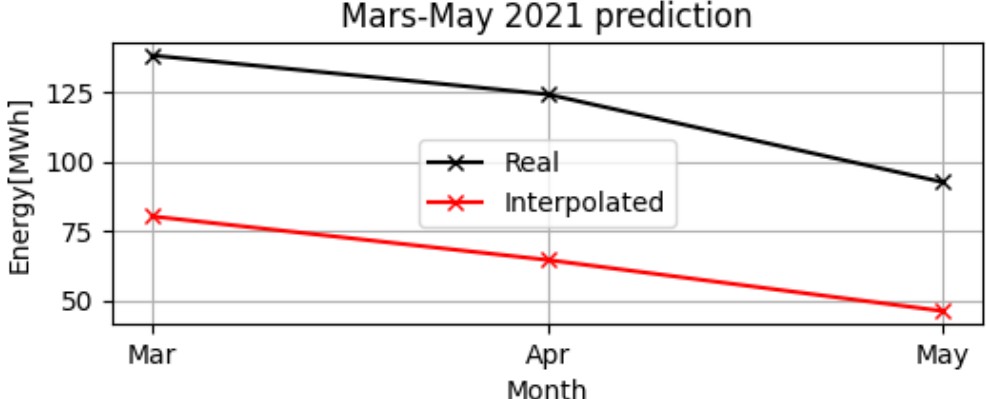

**Figure 18.** Predicted model and real electrical consumption for Months March–May in 2021 for UDJG.

Table 2 shows the total annual electrical consumption of UDJG in the three studied years and the relative percentages of consumption fluctuation when they are compared to those in 2019. The year 2020 was an academic year with online education only, which led, at UDJG, to a reduction in the electrical consumption by 36%. It is noteworthy that in 2021 there was an increase in the consumption by 30.63%, due, on one hand, to the partial resumption of face-to-face teaching and research activities, therefore, meaning that there was an increase in the number of students accommodated in the dormitories, and on the other hand, due to the modernization works that were carried out on the UDJG university campus.

From the data that are presented in Section 2, in 2019—before the pandemic—in UDJG, 4000 computers operated with an average consumption of 130 Wh. During the pandemic period of the 2020 academic year, when during the entire academic year, the didactic activity took place online, 1000 computers were operating with 3000 fewer units. On the UDJG campus, the students and researchers had about 3000 computers. Therefore, the total number of computers in UDJG for 2019 was 7000 units. Taking into account an average operation frequency of 4 h/day and 185 days per year, this results in a total annual electrical consumption of 673.4 MWh in 2019, representing 60.5% of the total consumption. In 2020, during the pandemic, due to the on-line teaching activities, and a small number of students or researchers being present on the campus, the number of computers had been reduced to 250 units. Thus, the electrical consumption for 2020 is 24.05 MWh. From the total electrical consumption of UDJG (Table 2), the consumption that is due to the operation of computers represents 60.5% in 2019 and 3.37% in 2020, respectively, as is presented synthetically in Table 3. If 2019 is taken into account as a reference year of the total annual consumption due to the use of computer units, then, in the pandemic period of 2020 the computers' electrical consumption decreased by 57.13%. In 2021, the teaching activities took place both online and face-to-face. Thus, the electrical consumption of the computer units for this year was not taken into account.

Based on a nonpublic survey that was conducted on a significant number of teachers and students of UDJG whose household energy footprints in 2020 did not change substantially when compared to those of 2019, the collected data showed an average monthly

electrical consumption of 125 kWh/month in 2019 and of 141 kWh/month in 2020. It was also showed that when compared to an average of 1 h/day before the pandemic in 2019, the computer activity in the homes of teachers and students increased to an average of 9 h/day during the pandemic in 2020, i.e., a difference of 8 h/day. Taking into account 300 days/year, for a 130 Wh computer, this results in a higher electrical consumption during 2020 when compared to that of 2019, by 312 kWh/year or by 26 kWh/month.

Although previous articles have used different methods of analyzing the real data, the relative values of the decrease in the electrical consumption during the 2020 pandemic compared to the normal period of academic education in 2019 can be assessed in the same way. Thus, in Table 4 are presented the percentages of electrical consumption decreases from 2020 when they are compared to those of 2019, which were reported in various articles and in the present study.

**Table 4.** Percentages of electrical consumption decrease in universities during pandemic period in 2020 when compared to those of 2019 that have been reported in several articles.

| Article | Percentage of Electrical Consumption Decrease in Universities during Pandemic Period in 2020 Compared to 2019 (%) |
|---|---|
| Reference citation number [15] | 33.2% |
| Reference citation number [16] | 20% |
| Reference citation number [19] | 25% |
| Reference citation number [17] | 46.61% |
| UVT—present study | 36.29% |
| UPB—present study | 20.6% |
| UPIT—present study | 26.41% |
| UDJG—present study | 20.6% |

As seen in Table 4, the values that have been calculated in the present study are close to the other values, which means that the pandemic period of 2020 was experienced in the same way regarding the reduction in the amount of electrical consumption in many universities around the world.

The analysis of the results that have been presented in previous articles, respectively in this study, indicates that the electrical consumption for the residential sector increased during the lockdown period in 2020 when these data are compared to those of the pre-pandemic 2019 period. The reported and collected data from the residential consumers denote that the majority of the increases in residential electrical consumption are due to the use of energy systems such as HVAC (heating and air-conditioned systems), appliances and lighting. At the same time, the intensive use of computers, sometimes in the number of 2–3 in the same house, has also led to a significant increase in the amount of electrical consumption. In this respect, Table 5 presents the percentages of the residential electrical consumption increases from 2020 when they are compared to those of 2019, which were reported in various articles and in the present study.

**Table 5.** Percentages of residential electrical consumption increase during pandemic period in 2020 when compared to those of 2019 that have been reported in several articles.

| Article | Percentage of Residential Electrical Consumption Increase during Pandemic Period in 2020 Compared to 2019 (%) |
|---|---|
| Reference citation number [13] | 19.3% |
| Reference citation number [20] | 23.4% |
| Reference citation number [21] | 30% |
| UVT—present study | 15.3% |
| UPB—present study | 19.5% |
| UPIT—present study | 12.5% |
| UDJG—present study | 12.8% |

As is seen in Table 5, the reported values in the present study are close to the other values, which means that the pandemic period of 2020 was experienced in the same way regarding the increases in residential electrical consumption around the world. Also, in [21]—line three of Table 5—the calculated percentage value of 30% had resulted from the intensive use of HVAC systems in residential housing in the USA during the pandemic period in 2020.

## 4. Conclusions

One of the most affected institutions during the pandemic period were universities, where major changes have occurred both in the new methods of online education as well as in the patterns of electrical consumption for universities and students' and teachers' residential housing.

The analysis of the data regarding the electrical consumption from the four Romanian universities leads to the highlighting of the decrease, in relative values, between 20.6% (UPB) and 36.21% (UVT) during the pandemic period, especially in 2020, compared to those of the pre-pandemic period in 2019. In addition, this study proposes the use of numerical approaches to define the impacts of the pandemic on the electrical consumption of universities. By applying the "polyfit" function to the interpolation polynomials of grade 9 with a very good accuracy, values of the coefficient $R^2$ that were higher than 0.9 were obtained. In one case, at UPB in 2021, the coefficient $R^2$ was low, 0.5931, a value that is explained by the very high electrical consumption during the summer (June–August) when, in the holiday period, a complex series of works and the modernization of the equipment were performed. This unregulated electrical consumption is not found in the consumption curve of the 2019 summer data.

Because, during the months of March–May in 2020 and 2021, online education took place in the studied Romanian universities, a prediction polynomial model of the electrical consumption was created using the "polyfit" function of the NumPy library for March–May in 2020 and it was compared with the real data of the same period in 2021. The metric distances between real and predicted values of electrical consumption of this period depended on the management of each university, because during this period, multiple modernizations, repairs and maintenances works were carried out.

Moreover, in this article, one calculated for each university the evolution of the electrical consumption percentage due to the use of computers during the pre- and pandemic periods. Using the data from the normal academic year in 2019, the consumption that was seen due to the use of computers represents an important percentage, between 11.28% (UPB) and 60.5% (UDJG), of the total electrical consumption of the studied universities. This electrical consumption is substantially reduced during 2020, with percentages that are between 75% (UVT) and 96.42% (UDJG). Based on the individual studies that were performed for several teachers and students, one estimated the average increase of residential electrical consumption during the pandemic period in 2020 to be about 20 kWh per month per computer/laptop unit, representing the transfer of a part of the energy consumption from the university to the household.

Considering that these numerical models and their analysis are valid for the study of any university in the world during the pre- and pandemic periods, it is important to know what the costs are for each part that is involved in the educational process in order to have techniques for predicting these costs and to negotiate the rights of those who offer educational services in such conditions. Based on the analysis that was conducted for this study, in order to decrease the electrical consumption of a university, it is recommended to implement procedures and systems for the turning off and leaving on of all types of electrical equipment.

Regarding the topics that have been presented in this article, the authors consider that they have highlighted all the important aspects that are related to the electrical consumption of the four universities in Romania during the pre- and pandemic periods. Also, the authors believe that the obtained results and the comments that are made in this article are

useful because this study can be a starting point for assessing the forecast of the electrical consumption of other institutions and managing scenarios in case of an event such as the pandemic.

**Author Contributions:** Conceptualization, H.A., N.B. and M.G.; methodology, P.C.A., M.S., E.D. and A.G.M.; software, I.C., H.A., L.M.I. and M.G.; validation, P.C.A., H.A., N.B. and M.G.; formal analysis, H.A.; investigation, M.S., E.D., A.G.M. and M.G.; resources, P.C.A., E.D., L.M.I. and M.G.; data curation, L.M.I.; writing—original draft preparation, P.C.A., H.A., A.G.M. and M.G.; writing—review and editing, M.S., E.D., N.B. and M.G.; visualization, M.S. and H.A.; supervision, P.C.A., I.C. and L.M.I. All authors have read and agreed to the published version of the manuscript.

**Funding:** This research received no external funding.

**Institutional Review Board Statement:** Not applicable.

**Informed Consent Statement:** Not applicable.

**Data Availability Statement:** Not applicable.

**Acknowledgments:** The authors thank the administrative services of their universities for the technical data provided.

**Conflicts of Interest:** The authors declare no conflict of interest.

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
