# Peer review of "Comparative and Predictive Analysis of Electrical Consumption during Pre- and Pandemic Periods: Case Study for Romanian Universities"

_sustainability, doi:10.3390/su141811346_

Round 1
Reviewer 1 Report (Previous Reviewer 3)
There are a lot of faults to be addressed in the paper, as below.
1. The content of the abstract does not show the simplicity to concentrate on the main point in the abstract. The authors must think of how to decrease the paper's redundancy.
2. Writing skill is still poor, such as in line 38: university consumption and line 39: residential consumption. Both subjects may confuse the reader.
3. Line 88: decrease in CO2 emissions; carbon dioxide needs to be a subscript.
4. Introduction: Although the authors try to explain the relation with electronic consumption during COVID 19. However, the paragraph indicates the divergency of the reading and is defocused after reading. This section has to downsize the content and focus on the electrical consumption during the pandemics.
5. The paper shall rewrite the introduction and hence the literature review.
6. The state of table 1 exhibits the Geographic distribution of universities and computers in Romania. That is not a precious description. Suggest you change to Geographic distribution of universities and the number of computers in Romania.
7. In the table, what is the meaning of the total? This is a confusing word.
8. Line 212, please specify how you take in-situ measurements.
9. Materials and Methods: Please downsize and simplify the content that the methodology concentrates on the electrical consumption. The authors shall pay attention to line 384~line 414 and enhance the methodology. The other part may revise as short as possible. Remember, the section is Materials and Methods, but we do not see your methodology architecture detailed.
10. Results and discussion: Line 416~line 432 shall move to Section 2: Materials and Methods. The authors shall then revise it a little.
11. In figures 5, 8, 9, 11, 13, and 17, the prediction of actual electrical consumption compared with polynomial model consumption is not consistent with each other. However, the other figures might be a look-good tendency. Could the authors make an explicit description?
12. Figure 5; what is the horizontal axis? Month? This is very strange to show 3.00, 3.25, 3.50........5.00. The indication is the basic of the month—March, April, May, June, July, August, etc. Please also check if the other figures are similar to the problems.
13. The technical part of the description in the figures is impoverished. How does the poly fit be made? How is the R2 estimated, and what is the graphic chart plotting in the estimation?
Author Response
AUTHORS’ RESPONSES TO THE REVIEWER 1
The authors thank the Reviewer 1 for their time and specific comments. We have studied all the comments and suggestions carefully and have made improvements which we hope to meet with approval. We believe that the paper has been further improved by addressing the Reviewer’s remarks.
The responses to the Reviewer’s comments are reported below. For the sake of clarity, the original comments are written in italic.
We are submitting the revised paper in which the added text is written in green color, for re-review.
- The content of the abstract does not show the simplicity to concentrate on the main point in the abstract. The authors must think of how to decrease the paper's redundancy
Response 1: Thanks for those useful suggestions on the manuscript. We made corrections and we rewrote the Abstract.
- Writing skill is still poor, such as in line 38: university consumption and line 39: residential consumption. Both subjects may confuse the reader.
Response 2: Thanks for those useful suggestions on the manuscript. We made corrections and we improved the writing skills for the whole article.
- Line 88: decrease in CO2 emissions; carbon dioxide needs to be a subscript.
Response 3: Thanks for those useful suggestions on the manuscript. We removed line 88.
- Introduction: Although the authors try to explain the relation with electronic consumption
during COVID 19. However, the paragraph indicates the divergency of the reading and is
defocused after reading. This section has to downsize the content and focus on the electrical
consumption during the pandemics.
Response 4: Thanks for those professional comments. We do agree with this point of view. We made corrections in the Introduction section and we focused on the electrical consumption during COVID 19.
- The paper shall rewrite the introduction and hence the literature review.
Response 5: Thanks for those useful comments. We rewrote the Introduction section and we improved the literature review accordingly.
- The state of table 1 exhibits the Geographic distribution of universities and computers in
Romania. That is not a precious description. Suggest you change to Geographic distribution of universities and the number of computers in Romania.
Response 6: Thanks for those useful suggestions on the manuscript. We removed Table 1.
- In the table, what is the meaning of the total? This is a confusing word.
Response 7: Thanks for those useful suggestions on the manuscript. We removed Table 1.
- Line 212, please specify how you take in-situ measurements.
Response 8: Thanks for those useful suggestions on the manuscript. Since the number of pages of the article would have increased unnecessarily by introducing 4 more figures to present the data acquisition systems of electrical consumption of the studied universities, I have inserted the following phrase: “Also, a comparative analysis of the monthly electrical consumption in pre- and pandemic period of the studied universities, located in the capital and in other three important cities, is performed, based on the data collected from monthly measurements.”
- Materials and Methods: Please downsize and simplify the content that the methodology concentrates on the electrical consumption. The authors shall pay attention to line 384~line 414 and enhance the methodology. The other part may revise as short as possible. Remember, the section is Materials and Methods, but we do not see your methodology architecture detailed.
Response 9: Thanks for those professional comments. We do agree with this point of view. We made corrections in the Materials and Methods section and we focused on the electrical consumption. We enhance the methodology description (line 384~line 414) and we introduced a methodology flowchart (Figure 2, page 8 in the revised version of article).
- Results and discussion: Line 416~line 432 shall move to Section 2: Materials and Methods. The authors shall then revise it a little.
Response 10: Thanks for those useful suggestions on the manuscript. We moved line 416~line 432 to Section 2. Also we revised and improved the whole Results and Discussion section. Also the Conclusion section is improved.
- In figures 5, 8, 9, 11, 13, and 17, the prediction of actual electrical consumption compared with polynomial model consumption is not consistent with each other. However, the other figures might be a look-good tendency. Could the authors make an explicit description?
Response 11: Thanks for those professional comments. We replaced all figures 2-17 and we made explicit descriptions of each one.
- Figure 5; what is the horizontal axis? Month? This is very strange to show 3.00, 3.25, 3.50........5.00. The indication is the basic of the month—March, April, May, June, July, August, etc. Please also check if the other figures are similar to the problems.
Response 12: Thanks for those useful suggestions on the manuscript. We replaced figures 5,9,13 and 17 and we made corrections of the horizontal axis accordingly.
- The technical part of the description in the figures is impoverished. How does the poly fit be made? How is the R2 estimated, and what is the graphic chart plotting in the estimation?
Response 13: Thanks for those professional comments. We improved the technical description of each figure and we made a detailed description of each step of numerical method. For example in step 2 (line 380 ~ line 386 in the revised version of article) we made an explicit description of R2: “The approximation polynomials are selected and validated if R-square (R2) values are bigger than 0.9. Graphical characteristics which put in evidence the real and modeled data of electrical consumption and the R2 values are presented.” Also, after this we specified (line 419 ~ line 421 in the revised version of article): “For determining with the good accuracy the polynomial approximations of electrical consumption for each university a Python script was developed to compute the R2.”
Sincerely,
Horia Andrei & 8 co-authors

Reviewer 2 Report (Previous Reviewer 2)
Manuscript title: Comparative and predictive analysis of electrical consumption during pre- and pandemic period. Case study for Romanian Universities
Comments:
1. Abstract is too long. The abstract contains a summary of the entire paper and can be up to 200 words long with only one paragraph.
2. Please add a flowchart for the methodology section.
3. The Prediction accuracy metric must be presented in the methodology section, not in the result section, please check the structure of these articles "Data mining with 12 machine learning algorithms for predict costs and carbon dioxide emission in integrated energy-water optimization model in buildings" and "A Hybrid Model with Applying Machine Learning Algorithms and Optimization Model to Forecast Greenhouse Gas Emissions with Energy Market Data."
4. Please clarify why authors don't use the other metrics for prediction accuracy evaluation?
5. Discussion section has poorly been written. Please discuss the results in a better manner, by comparing with the literature.
6. The structure of table 2 is not appropriate. Please check it.
Author Response
AUTHORS’ RESPONSES TO THE REVIEWER 2
The authors thank the Reviewer 2 for their time and specific comments. We have studied all the comments and suggestions carefully and have made improvements which we hope to meet with approval. We believe that the paper has been further improved by addressing the Reviewer’s remarks.
The responses to the Reviewer’s comments are reported below. For the sake of clarity, the original comments are written in italic.
We are submitting the revised paper in which the added text is written in green color, for re-review.
- Abstract is too long. The abstract contains a summary of the entire paper and can be up to 200 words long with only one paragraph.
Response 1: Thanks for those useful suggestions on the manuscript. We made corrections and we rewrote the Abstract accordingly.
2. Please add a flowchart for the methodology section.
Response 2: Thanks for those useful suggestions on the manuscript. We added a methodology flowchart (Figure 2, page 8 in the revised version of article) in Materials and Methods section.
- The Prediction accuracy metric must be presented in the methodology section, not in the result section, please check the structure of these articles "Data mining with 12 machine learning algorithms for predict costs and carbon dioxide emission in integrated energy-water optimization model in buildings" and "A Hybrid Model with Applying Machine Learning Algorithms and Optimization Model to Forecast Greenhouse Gas Emissions with Energy Market Data."
Response 3: Thanks for those professional comments. We introduced the prediction accuracy metric in the Materials and Methods section. Also we checked the two indicates articles and we introduced in the literature review (References section).
- Please clarify why authors don't use the other metrics for prediction accuracy evaluation?
Response 4: Thanks for those professional comments. In the last part of Section 2 (line 403 ~ line 411 in the revised version of article) we specified why the authors don’t use other metrics for prediction accuracy evaluation: “In previous studies [39] and [40] the authors have used machine learning algorithms as SVM, ANM, LASSO, LR, GB or RF for predict cost and gas emissions in integrated energy-water optimization model in buildings respectively to forecast greenhouse gas emissions. Based on large data collection the proposed algorithms provided results with an accurate prediction. Unlike these articles, the current study is based on a small number of electrical consumption data from each university, only 12 values for each year. Therefore, polynomial regression was chosen for the mathematical modeling of the annual electrical consumption of each university, which finally provided very precise results.”
- Discussion section has poorly been written. Please discuss the results in a better manner, by comparing with the literature.
Response 5: Thanks for those useful suggestions on the manuscript. We rewrote the Discussion section and we added the Table 3 (page 14 in the revised version of article) to put in evidence the electrical consumption due to the computers respectively the Table 5 (page 23 in the revised version of article) in order to present our results in a better manner, by comparing with results reported in literature. Also the Conclusion section is improved.
- The structure of table 2 is not appropriate. Please check it.
Response 6: Thanks for those useful suggestions on the manuscript. The Table 2 (renumbered as Table 1 in page 12 of the revised version of article) is restructured accordingly.
Sincerely,
Horia Andrei & 8 co-authors

Reviewer 3 Report (Previous Reviewer 1)
Dear Authors,
I would like to thank the author/s for their effort.
After careful revision, Kindly know that authors respond to each comment that I given to them.
For that I accept the paper in its current version of submission.
By the end, I want to say that you did a great job.
Thank you for your efforts.
Best Wishes
Author Response
AUTHORS’ RESPONSES TO THE REVIEWER 3
The authors thank the Reviewer 3 for their time and specific comments. We have studied all the comments and suggestions carefully and have made improvements which we hope to meet with approval. We believe that the paper has been further improved by addressing the Reviewer’s remarks.
The responses to the Reviewer’s comments are reported below. For the sake of clarity, the original comments are written in italic.
We are submitting the revised paper in which the added text is written in green color, for re-review.
Dear Authors,
I would like to thank the author/s for their effort.
After careful revision, kindly know that authors respond to each comment that I given to them.
For that I accept the paper in its current version of submission.
By the end, I want to say that you did a great job.
Thank you for your efforts.
Best Wishes
Response: We thank Reviewer 3 for the positive evaluations on our article and for the nice words addressed to the authors.
Best regards,
Horia Andrei & 8co-authors

Round 2
Reviewer 1 Report (Previous Reviewer 3)
The paper has shown much improvement except in the conclusion. The authors shall reduce the conclusion paragraph to as short as the abstract.
Author Response
AUTHORS’ RESPONSES TO THE REVIEWER 1
The authors thank the Reviewer 1 for their time and specific comments. We have studied all the comments and suggestions carefully and have made improvements which we hope to meet with approval. We believe that the paper has been further improved by addressing the Reviewer’s remarks.
The responses to the Reviewer’s comments are reported below. For the sake of clarity, the original comments are written in italic.
We are submitting the revised paper in which the added text is written in green color, for re-review.
The paper has shown much improvement except in the conclusion. The authors shall reduce the conclusion paragraph to as short as the abstract.
Response: Thanks for those useful suggestions on the manuscript. We made the required corrections and reduced the Conclusion paragraph by 50%, from 1136 words to 568 words.
Sincerely,
Horia Andrei & 8 co-authors

This manuscript is a resubmission of an earlier submission. The following is a list of the peer review reports and author responses from that submission.
Round 1
Reviewer 1 Report
Dear Authors,
I would like to thank the author/s for their effort.
A significant effort has been made by the authors to explore and propose a method for predicting, with good approximations, consumption in the case of online education all that in the matter of electric energy consumption during the pandemic. But there are some minor comments that need from authors to modify it to enhance the research:
1-In abstract, graphical abstract is requested to explain more what you will do in this article. Also, I prefer that the authors can add briefly part of finding.
2- Section (2) is almost three pages long without any references. It would be helpful if the author could add references to different universities.
3- I need a proof and explanation of the last sentence of section (3), page (7): "It was automatically observed that the data from each measurement system were taken from the tables, constituting the input data for the developed software applications.". Furthermore, before section (3), it would be better if the authors could describe the process of calculating the model numerically for each university with PyCharm Community.
4-The References in this article need a revision and rewrite it in the journal format specially the online references.
By the end, I want to say that you did a great job and all this comments is to encourage the author to enhance the quality of paper.
Thank you for your efforts.
Best Wishes
Reviewer 2 Report
Overall this writing of this paper is not mature enough to submit to a journal. First of all, there are many missing spaces in between sentences and typos, etc. The motivation and rationale to support the proposed model for forecast energy consumption are not clear. It is very difficult to see the research merit and contribution.
Comments:
1. Abstract should have one sentence per each: context and background, motivation, hypothesis, methods, results, conclusions. While the author presents the Abstract, answer the questions carefully: What problem did you study, and why is it important? What methods did you use? What were your main results? And what conclusions can you draw from your results? Please make your abstract with more specific and quantitative results while it suits broader audiences.
2. The originality of the paper needs to be stated clearly. It is importance to have sufficient results to justify the novelty of a high-quality journal paper. The Introduction should make a compelling case for why the study is useful along with a clear statement of its novelty or originality by providing relevant information and providing answers to basic questions such as: What is already known in the open literature? What is missing (i.e., research gaps)? What needs to be done, why and how? Clear statements of the novelty of the work should also appear briefly in the Abstract and Conclusions sections.
3. The current literature review is poor, the authors just list some previous studies one by one. You should further classify, summarize, and discuss the merit and demerit of these existing studies. This is helpful for identify your research questions.
4. Avoid lumping references as in "((Hinton et al., 2006), (Hinton & Salakhutdinov, 2006), (Luo et al., 2014), (Zeng et al., 2013), (L.-C. Chen et al., 2017))." and all other. Instead, summarize the main contribution of each referenced paper in a separate sentence.
5. Please clearly indicate why the proposed model is novel. The authors are recommended to use a table to compare the features of the proposed model with the previous similar models.
6. In In the Material and Methods section, (1. Please provide a better problem definition. 2. Please point out the main contributions of this paper. 3. What are the main assumptions of your model? 4. Adding a flowchart showing the step-by-step procure of the proposed study is recommended.)
7. Which algorithm is used to forecast energy consumption in the university? Which metrics are used to evaluate prediction accuracy?
Which data is used, and how is the data processing for the forecasting?
8. Result Section, 1. There is a need to add more discussion comments about the current state of the proposed study, types, and uses. 2. The main advantages and disadvantages of the proposed subject should be added.
9. As a key part of a paper, the Discussion should show the readers at least two elements: "breadth" and "depth". "Breadth" reflects whether the analytical results can be explained via different approaches. "Depth" reflects whether the analytical results completely answer the questions raised in the Introduction. My first sense shows the current Discussion is without enough insight. This should explore the significance of the results of the work, not repeat them.
10. Where is the Conclusion ?!!!
Reviewer 3 Report
I found the paper not a professional article for the journal. Also, the writing skill and composition of the architecture in the article are inferior, and many sentences are missing meaning. Thus, I suggest the paper needs consultation with a native English person to proceed with the amending and needs a rewrite of the technical paper to resubmit the journal. On the other hand, the portion indicates the problem as below.
1. The abstract does not show the research's main point and indicates the article's features.
2. The authors shall preciously address each noun in Tables 1-2 and rearrange the simplified table.
3. Table 3 must rearrange the table. This data seems a mess, and the reader cannot identify it.
4. Table 5 is the same as Table 3.
5. Table 7 is the same as Table 3.
6. Table 9 is the same as Table 3.
7. The conclusion needs to be downsizing and making a decisive contribution.
In the end, the reviewer does not see sufficient research and study of the technology for the innovation and pioneer.